# Grammar-Aligned Decoding

**Kanghee Park**[1][*]  **Jiayu Wang**[1][*]  **Taylor Berg-Kirkpatrick**[2]
**Nadia Polikarpova**[2]  **Loris D'Antoni**[1]
[1]University of Wisconsin-Madison  [2]University of California San Diego
{kpark247, jwang2782, ldantoni}@wisc.edu, {tberg, npolikarpova}@ucsd.edu

## Abstract

Large Language Models (LLMs) struggle with reliably generating highly structured outputs, such as program code, mathematical formulas, or well-formed markup. Constrained decoding approaches mitigate this problem by greedily restricting what tokens an LLM can output at each step to guarantee that the output matches a given constraint. Specifically, in *grammar-constrained decoding* (GCD), the LLM's output must follow a given grammar. In this paper we demonstrate that GCD techniques (and in general constrained decoding techniques) can *distort the LLM's distribution*, leading to outputs that are grammatical but appear with likelihoods that are not proportional to the ones given by the LLM, and so ultimately are low-quality. We call the problem of aligning sampling with a grammar constraint, *grammar-aligned decoding* (GAD), and propose *adaptive sampling with approximate expected futures* (ASAp), a decoding algorithm that guarantees the output to be grammatical while provably producing outputs that match the conditional probability of the LLM's distribution conditioned on the given grammar constraint. Our algorithm uses prior sample outputs to soundly overapproximate the future grammaticality of different output prefixes. Our evaluation on code generation and structured NLP tasks shows how ASAp often produces outputs with higher likelihood (according to the LLM's distribution) than existing GCD techniques, while still enforcing the desired grammatical constraints. [2]

## 1  Introduction

Despite their remarkable success, pre-trained Large Language Models (LLMs) often struggle with generating highly structured outputs, such as program code, configuration files, or mathematical formulas. A naïve approach to enforcing structure is *rejection sampling*, which repeatedly samples strings from the LLM and checks them against a validity oracle, typically in the form of a *context-free grammar* (CFG). Rejection sampling is highly inefficient or simply intractable for restrictive grammars and long output sequences—i.e., most generated strings will not be in the target grammar.

Constrained decoding addresses the inefficiency of rejection sampling by greedily "forcing" the LLM output to satisfy the given constraint. Specifically, when the constraint is given as a grammar, *grammar-constrained decoding* (GCD) [7, 27, 28], can build automata that allow for on-the-fly masking of tokens that will provably lead to outputs outside of the grammar during decoding.

While GCD does not incur the overhead of rejection sampling—i.e., the generated output is always in the language of the grammar—we show that GCD and in general all forms of structured decoding introduce a new problem: **structured decoding distorts the LLM's learned language distribution**, effectively hindering the LLM's capabilities.

---

[*]Equal contribution

[2]Our code, datasets, and checkpoints are available at: `https://github.com/ebmoon/transformers-GAD`.

This paper introduces and formalizes *grammar-aligned decoding* (GAD), the problem of sampling from an LLM so that the outputs (1) are guaranteed to adhere to a given grammar, and (2) are unbiased *wrt.* the LLM's distribution. Although exact GAD is intractable in general (similar to rejection sampling), we propose a new adaptive decoding algorithm for *approximate* GAD, which starts off as GCD and gradually converges to the LLM's distribution, and thus allows trading off between efficiency and accuracy. The algorithm, which we dub *Adaptive Sampling with Approximate Expected Futures* (ASAp), is built "on top" of existing constrained decoding algorithms. Whereas GCD approaches simply mask out tokens that lead to non-grammatical sequences for a given prefix, ASAp remembers for all sampled prefixes the probability associated with masked-out tokens and uses it to upper bound the probability of grammaticality. By updating this bound when more samples are observed, the decoding algorithm converges to the desired probability distribution—i.e., it samples outputs from the LLM-induced probability conditioned on the outputs being accepted by the grammar. The idea works for any structured decoding approach and not just for GCD, but in this paper we focus our evaluation on constraints expressed via grammars.

We evaluate ASAp on two structured prediction tasks: formal program synthesis and constituency parsing. Our experiments on program synthesis and NLP tasks show that GCD techniques generate outputs that are grammatical but unlikely according to the LLM, while with ASAp, the likelihood of the generated outputs improves over time, converging to the target constrained LLM—i.e., GAD better respects the LLM while still enforcing the constraints.

## 2 Grammar-Aligned Decoding

In this section, we formalize the problem of *grammar-aligned decoding* (GAD) as decoding from an autoregressive language model while enforcing the output sequence to be accepted by a given context-free grammar. We also demonstrate the limitations of existing approaches to this problem.

**Language Models**   An (autoregressive) language model defines a probability distribution $P$ on the set of all strings $w \in \Sigma^*$ over a vocabulary of tokens $\Sigma$ via a product of left-to-right next-token conditional distributions $P(w_1 \ldots w_n) = \Pi_{i=1}^n P(w_i \mid w_{1:i-1})$.

**Context-Free Grammars**   A *context-free grammar* (CFG) is a quadruple $\mathcal{G} = (\Sigma, \mathcal{N}, S, \mathcal{R})$, where $\Sigma$ is a vocabulary of tokens (also called terminal symbols), $\mathcal{N}$ is a finite set of non-terminal symbols, $S \in \mathcal{N}$ is the starting non-terminal, and $\mathcal{R}$ is the set of production rules. An example CFG is shown in Fig. 1. A grammar $\mathcal{G}$ defines a *single-step derivation* relation on sequences of symbols $\alpha, \beta, \gamma \in (\mathcal{N} \cup \Sigma)^*$: $\alpha A \gamma \Rightarrow \alpha \beta \gamma$ if $A \to \beta \in \mathcal{R}$. The reflexive transitive closure of this relation is called *derivation* and written $\Rightarrow^*$. A sequence of tokens $w$ is a *sentence* if it is derivable from $S$; the set of all sentences is called the *language* of the grammar $\mathcal{G}$, that is, $\mathcal{L}(\mathcal{G}) = \{w \in \Sigma^* \mid S \Rightarrow^* w\}$. The following example illustrates these definitions.

$$
\begin{aligned}
S &::= \quad \texttt{00000} \mid \texttt{1}A_2 \\
A_i &::= \quad \texttt{0}A_{i+1} \mid \texttt{1}A_{i+1}, \text{ for } i = 2, 3, 4 \\
A_5 &::= \quad \texttt{0} \mid \texttt{1}
\end{aligned}
$$

Figure 1: CFG $\mathcal{G}_{sk}$ over tokens $\Sigma = \{0, 1\}$, written in Backus-Naur form (BNF) notation. This grammar accepts the string `00000` and all length-5 strings that start with a 1.

**Example 1** (CFG Derivations). *Given the CFG $\mathcal{G}_{sk}$ shown in Fig. 1, the string `00000` belongs to the language $\mathcal{L}(\mathcal{G}_{sk})$ because it can be derived using the derivation $S \Rightarrow$ `00000`. The string `10101` is also in $\mathcal{L}(\mathcal{G}_{sk})$ and can be derived as follows:*

$$S \Rightarrow \texttt{1}A_2 \Rightarrow \texttt{10}A_3 \Rightarrow \texttt{101}A_4 \Rightarrow \texttt{1010}A_5 \Rightarrow \texttt{10101}$$

*Each step replaces a nonterminal symbol using a production rule in $\mathcal{G}_{sk}$—e.g., in the string `10`$A_3$, the nonterminal $A_3$ is rewritten as `1`$A_4$ by applying the rule $A_3 \to \texttt{1}A_4$, resulting in the string `101`$A_4$.*

In addition, we define the *prefix language* of $\mathcal{G}$ as the set of all prefixes of sentences in $\mathcal{L}(\mathcal{G})$: $\mathcal{L}_{\text{prefix}}(\mathcal{G}) = \{w \in \Sigma^* \mid wv \in \mathcal{L}(\mathcal{G})\}$.

**Grammar-Aligned Decoding**   Given a model distribution $P$ and a CFG $\mathcal{G}$, *grammar-aligned decoding (GAD)* is the task of sampling from the distribution $Q^{P,\mathcal{G}}$ that is *proportional* to $P$ but *restricted* to sentences in $\mathcal{G}$:

$$Q^{P,\mathcal{G}}(w) = \frac{\mathbb{1}[w \in \mathcal{L}(\mathcal{G})] \cdot P(w)}{\sum_{w'} \mathbb{1}[w' \in \mathcal{L}(\mathcal{G})] \cdot P(w')}$$

When $P$ and $\mathcal{G}$ are clear from context, we will write $Q(w)$ instead of $Q^{P,\mathcal{G}}(w)$.

**Example 2** (GAD). *Consider the distribution $P$ that arises from prompting an LLM to "generate a binary string that ends with a 1". We expect $P$ to assign high probability to strings of the form $(0 \mid 1)^*1$—i.e. those that satisfy the prompt (`Mixtral-8x7B-Instruct-v0.1` (temperature=1) generates binary strings that end with a 1 around 90% of the time.) A snippet of a possible distribution $P$ is depicted in Fig. 2.*

*Suppose we constrain the model's output to the language of the grammar $\mathcal{G}_{sk}$ in Fig. 1, which only accepts strings of length 5. Moreover, $\mathcal{G}_{sk}$ only accepts one string that starts with 0, i.e., 00000, which does not end with 1. In Fig. 2, the grayed out parts of the trie are tokens that lead to sequences outside of the grammar $\mathcal{G}_{sk}$. According to the definition of GAD, the target sampling distribution $Q^{P,\mathcal{G}_{sk}}$ should assign: (i) high probability to all eight strings of the form $1w_2w_3w_41$—which conform both to the grammar and the prompt; (ii) low probability to the string 00000—which conforms to the grammar but not the prompt; and (iii) zero probability to all other strings.*

**Exact GAD** Can one exactly sample from $Q^{P,\mathcal{G}}$? Rejection sampling, which repeatedly draws from $P$ until a sample lands in $\mathcal{L}(\mathcal{G})$, provably yields exact samples according to $Q^{P,\mathcal{G}}$, but if $P$ assigns most of its mass outside of $\mathcal{L}(\mathcal{G})$, it is intractably slow, especially if the prompt is not including information about the grammar (see [27]). For Ex. 2, rejection sampling would be highly inefficient because the model would generate many strings that are not of length five.

In contrast, exact sampling from $P$ is efficient because its joint distribution is represented by a product of easily computed left-to-right conditionals, enabling ancestral sampling (i.e., generating tokens left to right, conditioned on already generated tokens). Can we similarly factor $Q$ into a product of left-to-right conditionals $Q^{P,\mathcal{G}}(w_i|w_{1:i-1})$, to enable ancestral sampling?

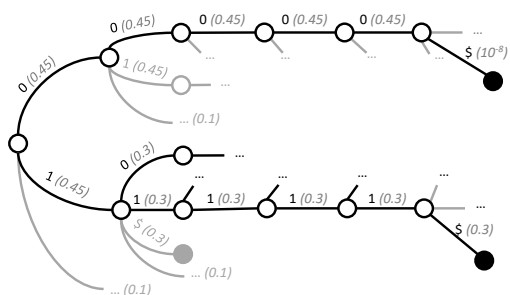

Figure 2: Fragment of the conditional model distribution $P$ for Ex. 2 depicted as a trie. Each node corresponds to a prefix $w_{1:i-1}$, and each edge is annotated with the next token $w_i$ and its conditional probability $P(w_i \mid w_{1:i-1})$. Filled nodes are complete strings. Grayed out parts of the trie are outside of the grammar $\mathcal{G}_{sk}$.

For simplicity, let us assume that $P$ is a distribution over sequences of exactly length $n$ (although, in practice, language models can produce 'stop' tokens which allow for a valid distribution on sequences of all lengths). The exact conditionals of $Q^{P,\mathcal{G}}$ are given by:

$$
\begin{aligned}
Q^{P,\mathcal{G}}(w_i \mid w_{1:i-1}) &\propto \sum_{w_{i+1:n}} \left[ \mathbb{1}[w \in \mathcal{L}(\mathcal{G})] \cdot \Pi_{j=i}^{n} P(w_j \mid w_{1:j-1}) \right] \\
&\propto P(w_i \mid w_{1:i-1}) \cdot \mathbb{E}_{P(w_{i+1:n}|w_{1:i})}[\mathbb{1}[w \in \mathcal{L}(\mathcal{G})]]
\end{aligned}
\tag{1}
$$

Thus, exact left-to-right sampling from $Q^{P,\mathcal{G}}$ consists of sampling from model conditionals $P(w_i \mid w_{1:i-1})$, with an additional weighting term $c(w_{1:i}) = \mathbb{E}_{P(w_{i+1:n}|w_{1:i})}[\mathbb{1}[w \in \mathcal{L}(\mathcal{G})]]$ that considers the grammar.

We refer to $c(w_{1:i})$ as *expected future grammaticality* (EFG), *i.e.* the probability that a continuation of $w_{1:i}$ sampled from $P$ lands in $\mathcal{L}(\mathcal{G})$. Using this notation, we can write the exact left-to-right sampling conditional explicitly as:

$$
Q^{P,\mathcal{G}}(w_i \mid w_{1:i-1}) = \frac{P(w_i \mid w_{1:i-1}) \cdot c(w_{1:i})}{\sum_{w_i'} P(w_i' \mid w_{1:i-1}) \cdot c(w_{1:i-1}, w_i')}
\tag{2}
$$

To see why computing this conditional is intractable, consider using dynamic programming to compute $c(w_{1:i})$ by marginalizing over a product of potential functions: the set of model conditionals and an indicator potential for the grammar. While the indicator potential can be factorized across rules in the grammar, the model's contribution generally does not factorize: in practice, the final conditional probability $P(w_n \mid w_{1:n-1})$ is a global potential function, defined by a non-linear neural network touching every variable. Thus, the main goal of this paper is to develop effective approximations to the EFG $c(w_{1:i})$, which would enable us to compute the left-to-right conditionals of $Q$.

**Limitations of Grammar-Constrained Decoding**   Existing work [27, 28] has proposed *grammar-constrained decoding* (GCD) as a way to efficiently sample from an autoregressive language model subject to grammar constraints. Although the exact details of these techniques vary depending on class of grammars they support, the common thread is that they rely on an *incremental parser*, which can efficiently check whether a given string $w$ is a prefix of a sentence in the grammar, i.e., $w \in \mathcal{L}_{\text{prefix}}(\mathcal{G})$. When given a sentence $w_{1:i-1}$, GCD techniques use this parser during decoding to mask out any next token $w_i$ that results in a prefix $w_{1:i}$ for which no completion will produce a sequence in the grammar. Using the trie in Fig. 2 as an example, one can think of GCD as sampling a path through the trie by selecting only among the black outgoing edges from every node, proportional to their conditional probabilities in the diagram (*e.g.* the first token is 0 or 1 with equal probability).

In terms of the GAD problem, we can view GCD as approximating the exact left-to-right conditionals $Q^{P,\mathcal{G}}(w_i \mid w_{1:i-1})$ by the conditional distribution $\tilde{Q}_{\text{GCD}}(w_i \mid w_{1:i-1})$, defined as follows:

$$\tilde{Q}_{\text{GCD}}(w_i \mid w_{1:i-1}) = \frac{P(w_i \mid w_{1:i-1}) \cdot \mathbb{1}[w_{1:i} \in \mathcal{L}_{\text{prefix}}(\mathcal{G})]}{\sum_{w_i'} P(w_i' \mid w_{1:i-1}) \cdot \mathbb{1}[w_{1:i-1}, w_i' \in \mathcal{L}_{\text{prefix}}(\mathcal{G})]}$$

Though not originally formulated in this way, we can view recent work on GCD [27, 28] as forming a binary approximation $\mathbb{1}[w_{1:i} \in \mathcal{L}_{\text{prefix}}(\mathcal{G})]$ to the EFG $c(w_{1:i})$. In other words, while GCD considers the *possibility* of future grammaticality, it makes no attempt to integrate the model's likelihood to estimate *expected* future grammaticality, which can lead to substantial bias in the sampling distribution—i.e., every EFG such that $c(w_{1:i}) > 0$ will simply be approximated via the value 1.

**Example 3** (GCD). *Consider again the GAD problem from Ex. 2, where our target sampling distribution $Q^{P,\mathcal{G}_{sk}}$ assigns high probability to strings that both start and end with a 1 and a low probability to the string 00000. However, we observe that GCD [8] generates strings ending with a 1 only 30% of the time—i.e., GCD has effectively ruined the LLM's ability to follow the prompt by biasing sampling towards 00000, an incorrect output.*

*When generating the first token (0 or 1), the GCD algorithm does not know how many grammatical strings can start with each character and, more importantly,* how likely *these strings are under $P$. Since both tokens 0 and 1 have the possibility of leading to a grammatical string, GCD will estimate their expected future grammaticality as 1, and choose each of them roughly half of the time (since $P(0) \approx P(1)$). Once GCD has chosen 0, however, it becomes "trapped" in the part of the search space where the only grammatical string is the low-probability sequence 00000.*

Ex. 3 illustrates how existing GCD approaches can hinder the language model's abilities to explore the space of possible outputs according to the learned distribution, thus highlighting the importance of designing a better approximation to the EFG $c(w_{1:i})$; this is addressed in the next section.

## 3   Adaptive Sampling with Approximate Expected Futures (ASAp)

In this section, we propose an adaptive sampling algorithm that iteratively builds better approximations of the future grammaticality of a sequence. Our procedure operates by sampling *repeatedly*, each time bounding lost probability mass to provably ungrammatical areas of the search space in order to better guide the next sampling iteration. As a result, our algorithm converges over many iterations to exact samples from the constrained LLM distribution, allowing for a flexible trade-off between efficiency and accuracy.

**Overview of the Algorithm**   GCD approaches poorly approximate the desired distribution because they greedily sample prefixes without worrying about the EFG. When sampling the first token in Ex. 3, GCD simply uses the likelihood for tokens 0 and 1 assigned by the LLM without considering the probability that these next tokens would result in grammatical completions if sampling were unconstrained—i.e. without incorporating the critical EFG re-weighting terms that are necessary for unbiased sampling from the constrained LLM distribution. However, if GCD ends up sampling 0 as the first token for Ex. 3, it will necessarily sample the string 00000 since no other sequences starting with 0 are allowed by the grammar. We can "learn" from this result: the true probability mass assigned to all grammatical sequences starting with a 0 is not 0.45 as the LLM's next token probability would have us believe; instead, the total grammatical mass in this section of the search space is the joint probability of the single string 00000, which is the much lower value of $0.45^5 * 10^{-8}$

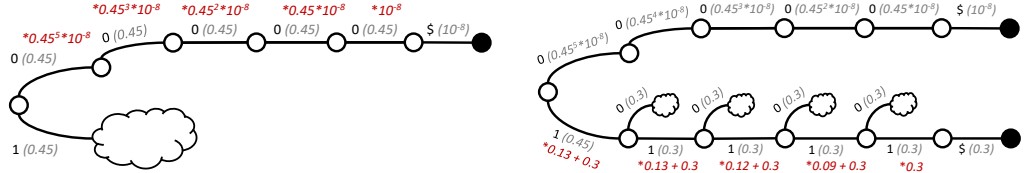

Figure 3: Illustration of the trie built by ASAp after sampling `00000` as the first string (left) and after sampling `11111` as the second string (right). EFG updates after each iteration are shown in red.

as depicted in Fig. 3. In other words, simply by sampling `00000`, we can better approximate (in this case, exactly) the EFG of tokens along this path.

The key insight behind our algorithm, which we call ASAp, is that we can iterate this process of discovering lost grammatical probability mass by repeatedly sampling and revising transition weights after each sample is produced. More formally, we can think of this procedure as starting with GCD's over-approximation to each EFG $c(w_{1:i})$ term, and then, through repeated sampling and discovery of mass assigned to non-grammatical completions, reducing each overapproximation to make it more accurate. In the limit, the approximations converge to exact EFG estimates and unbiased sampling.

Two possible first iterations of the ASAp algorithm are depicted in Fig. 3. In the first iteration (left of Fig. 3), after sampling the sequence `00000`, the algorithm directly addresses the issue that arose in Ex. 3 by attempting to better approximate the probability mass of potential grammatical completions of each prefix of `00000` (red quantities). For example, the expected future grammaticality of the prefix `0000` it is now $0.45 * 10^{-8}$—i.e., the algorithm effectively "looks ahead" to determine that only one valid (but low probability) string `0$` that can follow `0000`. The ideas developed in GCD allow us to efficiently compute, for a given string, the likelihood of the next tokens that will immediately result in non-grammaticality.

If we only sample one string from the LLM, we cannot hope to do better than GCD in terms of sampling faithfully in a grammar-aligned way. However, if we were to now sample once more, we could now better direct our sampling strategy. In the second iteration (right of Fig. 3), the string `11111` is sampled and the expected future grammaticality is updated (red quantities). Note that at this point the probabilites assigned to the string `00000` from the earlier iteration have already been updated.

By repeating the above approach multiple times (i.e., by producing more samples), the ASAp algorithm produces precise approximations of the expected future grammaticalities and thus better samples from the constrained LLM.

**Algorithm Formalization**    The key quantity that the algorithm approximates based on past samples is the *expected future grammaticality* (EFG) $c(w_{1:i}) = \mathbb{E}_{p(w_{i+1:n}|w_{1:i})}[\mathbb{1}[w \in \mathcal{L}(\mathcal{G})]]$. At iteration $m + 1$, our algorithm uses the set of samples $S = \{s_1, \ldots, s_m\}$ observed so far to compute an overapproximation $\tilde{c}_S(w_{1:i})$ of $c_S(w_{1:i})$ for every possible string $w_{1:i}$. The overapproximation is inductively defined:

$$\begin{array}{ll} \tilde{c}_S(w_{1:i}) = \mathbb{1}[w_{1:i} \in \mathcal{L}_{\text{prefix}}(\mathcal{G})] & \text{no string in } S \text{ starts with } w_{1:i} \\ \tilde{c}_S(w_{1:i}) = \sum_{w_{i+1}} P(w_{i+1} \mid w_{1:i}) \cdot \tilde{c}_S(w_{1:i+1}) & \text{otherwise} \end{array} \quad (3)$$

Intuitively, if no samples in $S$ start with the prefix $w_{1:i}$, then $\tilde{c}_S(w_{1:i})$, the overapproximation of EFG is simply whether the string is or is not a valid prefix in the grammar—i.e. the same overapproximation used by GCD. If, on the other hand, we *have* encountered the prefix $w_{1:i}$ before in previous samples in $S$, the overapproximation uses the next token likelihoods that were computed during the previous sampling runs of the algorithm to compute a better estimate of EFG.

For example, in Fig. 3, once we have sampled the sequences `00000` and `11111`, we have that $\tilde{c}_S(0000) = 0.45 * 10^{-8}$ and $\tilde{c}_S(110) = 1$ (i.e., we have not seen a sample with the prefix `110` yet).

**Theorem 1.** $\forall w_{1:i} \in \Sigma^*, \tilde{c}_S(w_{1:i}) \geq c(w_{1:i})$.

*Proof.* To see that $\tilde{c}_S(w_{1:i})$ is indeed an upperbound on $c(w_{1:i})$, consider two cases: First, suppose $w_{1:i}$ is not a prefix of any string in $S$. In this case, $\tilde{c}_S(w_{1:i}) = \mathbb{1}[w_{1:i} \in \mathcal{L}_{\text{prefix}}(\mathcal{G})]$ and, like GCD, provides a trivial upper bound. When $\mathbb{1}[w_{1:i} \in \mathcal{L}_{\text{prefix}}(\mathcal{G})] = 0$, there is

no possibility of grammaticality along this path and the EFG is therefore also zero. When $\mathbb{1}[w_{1:i} \in \mathcal{L}_{\text{prefix}}(\mathcal{G})] = 1$ it trivially bounds EFG, which is a probability. Second, we need to prove that $\forall w_{1:i} \in \text{prefix}(S), \tilde{c}_S(w_{1:i}) \geq c(w_{1:i})$. where $\text{prefix}(S)$ is the set of (finitely many) prefixes of string in $S$. We proceed by induction, where the base case is when $w_{1:i}$ is in $\text{prefix}(S)$ but no $w_{1:i+1}$ is in $\text{prefix}(S)$ for any possible next token $w_{i+1}$. Consequently, every $w_{1:i+1}$ falls under the first case, leading us to the following inequality:

$$\tilde{c}_S(w_{1:i}) = \sum_{w_{i+1}} P(w_{i+1} \mid w_{1:i}) \cdot \tilde{c}_S(w_{1:i+1}) \geq \sum_{w_{i+1}} P(w_{i+1} \mid w_{1:i}) \cdot c(w_{1:i+1}) = c(w_{1:i}) \quad (4)$$

Next, we move on to the inductive step where $w_{1:i}$ is in $\text{prefix}(S)$ and for any $w_{i+1}$, the string $w_{1:i+1}$ can either be a node that is not a prefix of $S$, which falls under the first case, or it can in $\text{prefix}(S)$, for which the property holds by induction. Therefore, the reasoning used in Eq. 4 works for the inductive case as well. $\qquad\square$

The sampling procedure itself proceeds autoregressively like GCD, but using the iteratively updated EFG estimates we have just defined, $\tilde{c}_S$. Specifically, the left-to-right sampling conditional for our procedure, $\tilde{Q}_S(w_i|w_{1:i-1})$, after having previously sampled the strings in $S$, is defined as follows:

$$\tilde{Q}_S(w_i|w_{1:i-1}) = \frac{P(w_i \mid w_{1:i-1}) \cdot \tilde{c}_S(w_{1:i})}{\sum_{w_i'} P(w_i' \mid w_{1:i-1}) \cdot \tilde{c}_S(w_{1:i-1}, w_i')} \quad (5)$$

Our overall algorithm, which is presented in Algorithm 1, then proceeds iteratively, using past samples to improve subsequent samples. Whenever the sample set $S$ is updated with a new sample $w_{1:n}$, the over-approximation $\tilde{c}$ is updated for the prefixes of $w_{1:n}$. The update begins at the end of the sequence and proceeds backward toward the start, by the recursive definition in Eq. 3. In the listing, we assume that we are only interested in the final sample, but in our evaluation we will analyze whether the algorithm induces the desired distribution.

---

**Algorithm 1** ASAp algorithm

---

Initialize $S := \{\}, \tilde{c}_S(\cdot) := 1$
**for** $m \leq M$ **do**
$\quad$ Draw $w_{1:n} \sim \tilde{Q}_S$ via ancestral sampling
$\quad$ $S := S \cup \{w_{1:n}\}$
$\quad$ **for** $i$ in $(n-1) \ldots 1$ **do**
$\quad\quad$ **for** $w'$ in $\{w' \mid w_{1:i} \cdot w' \notin \mathcal{L}_{\text{prefix}}(\mathcal{G})\}$ **do**
$\quad\quad\quad$ $\tilde{c}_S(w_{1:i} \cdot w') := 0$
$\quad\quad$ $\tilde{c}_S(w_{1:i}) := \sum_{w'} P(w' \mid w_{1:i}) \cdot \tilde{c}_S(w_{1:i} \cdot w')$
**return** Final sample $w_{1:n}$

---

Next we provide a proof that this algorithm converges to exact estimates of EFG in the limit of infinite iterations, and therefore to exact samples from the constrained LLM distribution. The theorem assumes almost sure termination of ancestral sampling in the unconstrained LLM distribution $P$—i.e., the LLM eventually terminates.

**Theorem 2.** *Assume that as $L \to \infty$, the distribution $P$ assigns vanishingly small probability mass to sequences longer than length $L$. Now, let $S_m = \{s_1, \ldots, s_m\}$ be the set of recorded samples up to the $m$th iteration of ASAp. Then, $\forall w_{1:i} \in \mathcal{L}_{\text{prefix}}(\mathcal{G}), \tilde{c}_{S_m}(w_{1:i}) \xrightarrow{p} c(w_{1:i})$ as $m \to \infty$.*

*Proof.* Let $w_{1:i}$ be an arbitrary sequence in $\mathcal{L}_{\text{prefix}}(\mathcal{G})$. The approximation gap after $m$ iterations of sampling with ASAp, $\epsilon_m = \tilde{c}_{S_m}(w_{1:i}) - c(w_{1:i})$, is equal to the marginal probability under $P$ of all *ungrammatical* continuations of $w_{1:i}$ that *have not yet been encountered* in the first $m$ samples, $S_m$. Now consider an arbitrarily small $\epsilon > 0$. By assumption, there exists an $L$ such that the probability mass $P$ places on sequences longer than $L$ is less than $\epsilon$. Further, ASAp samples according to $P$, but re-weighted by an upper bound on the true EFG (Theorem 1). Thus, the probability of encountering a *previously unseen* ungrammatical continuation of $w_{1:i}$ no longer than $L$ on any given iteration is at least as high as the probability of encountering the same continuation when sampling directly from $P$. Because the number of sequences no longer than $L$ is finite, this implies that the probability mass under $P$ of ungrammatical continuations of $w_{1:i}$ that are no longer than $L$ and that *are not yet encountered in $S_m$* becomes vanishingly small as as $m \to \infty$. The remaining unencountered ungrammatical continuations of $w_{1:i}$ are longer than $L$, and thus their total mass is bounded by $\epsilon$. Therefore $P(\epsilon_m > \epsilon) \to 0$ as $m \to \infty$.

$\qquad\square$

```
; Determines what terms can appear          ; Determines what terms can appear
(set-logic SLIA)                            (set-logic BV)
; The function to synthesize                ; The function to synthesize
(synth-fun f ((name String)) String         (synth-fun inv
    ; The grammar for f to be synthesized in     ((s (BitVec 4)) (t (BitVec 4)))
    ((Start String (S))                          (BitVec 4))
     (S String                              ; Helper functions
        (name "_" "."                       (define-fun min () (BitVec 4)
         (str.++ S S)                           (bvnot (bvlshr (bvnot #x0) #x1)))
         (str.at S I)                       (define-fun max () (BitVec 4) (bvnot min))
         (str.replace S S S)                (define-fun l
         (str.substr S I I)))                   ((s (BitVec 4)) (t (BitVec 4))) Bool
      (I Int                                    (bvsgt (bvnot (inv s t)) t))
         (0 1 2 (+ I I) (- I I)             (define-fun SC
          (str.len S)                           ((s (BitVec 4)) (t (BitVec 4))) Bool
          (str.indexof S S I)))))              (distinct t max))

; Specifications to satisfy                 ; Specifications to satisfy
(constraint (= (f "Nancy_FreeHafer") "N.F."))   ; with universally quantified variables
(constraint (= (f "Andrew_Cencici") "A.C."))    (declare-var s (BitVec 4))
(constraint (= (f "Jan_Kotas") "J.K."))         (declare-var t (BitVec 4))
(constraint (= (f "Mariya_Sergienko") "M.S."))  (constraint (=> (SC s t) (l s t)))

        (a) SLIA/initials-small                 (b) INV-BV/find_inv_bvsle_bvlshr1_4bit
```

```
Start ::=  S                               Start ::=  BV
    S ::=  name │ " " │ "."                    BV ::=  s │ t
        │  str.++ S S │ str.at S I                 │  #x0 │ #x7 │ #x8
        │  str.replace S S S                       │  bvneg BV │ bvnot BV
        │  str.substr S I I                        │  bvadd BV BV │ bvsub BV BV
    I ::=  0 │ 1 │ 2 │ + I I │ - I I              │  bvand BV BV │ bvlor BV BV
        │  str.len S │ str.indexof S S I           │  bvlshl BV BV │ bvlshr BV BV
         (c) Grammar for f                          (d) Grammar for inv
```

Figure 4: (a) A SLIA problem in which the grammar for the target function is explicitly defined. (b) INV-BV problem in which the grammar for the target function inv is implicitly defined. (c) The explicitly defined grammar for f written in BNF notation. (d) The implicitly defined grammar for inv written in BNF notation. The grammar is implicitly defined by primitives of BV logic and parameters of inv. The goal of each problem is to find an implementation for synth-fun functions that satisfies all the constraints within a specified grammar—i.e., to find implementation of f in the grammar (c) and inv in the grammar (d).

## 4 Experiments

We implemented the ASAp algorithm as an extension of the Transformers-CFG implementation of GCD [8]. When the LLM generates a sequence $w_{1:n}$, the ASAp algorithm keeps track of the original LLM's probability $P(w_i \mid w_{1:i-1})$ for $1 \leq i \leq n$ and the set of allowed next tokens $\{w_i \mid w_{1:i-1}, w_i' \in \mathcal{L}(\mathcal{G})\}$ determined by the incremental parser in the Transformers-CFG library. After the LLM finishes generating a sequence, our implementation of ASAp updates the overapproximation $\tilde{c}_S$ from the end of sequence by back-propagating the quantity 1 minus probability of the tokens that will for sure lead to non-grammatical sequences. The implementation of ASAp updates $\tilde{c}_S(w_{1:n-1}, w_n')$ for all possible tokens $w_n'$, and then moves on to update $\tilde{c}_S(w_{1:n-2}, w_{n-1}') \ldots, \tilde{c}_S(w_1, w_2'), \tilde{c}_S(w_1')$ using Equation (3).

**Datasets and Models.** We consider the benchmark from Example 3 and three structured-decoding tasks. Two of our tasks involve solving Syntax-Guided Synthesis Problems (SyGuS) [2]. SyGuS is a standardized format where one provides a logical specification and a context free grammar of first-order terms and the goal is to synthesize a term in the grammar that satisfies the specification. SyGuS is a natural fit for GAD and we consider two tasks from the standard SyGuS benchmarks where grammars vary from benchmark to benchmark: strings with linear integer arithmetic (SLIA) and loop invariant generation with bit-vector arithmetic (INV-BV). In the former, the grammar is used to restrict what constant strings one can use when building string-manipulating programs and in

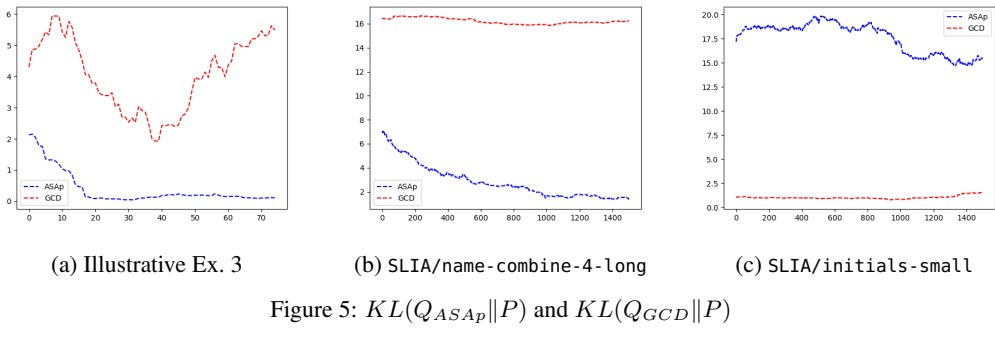

| (a) Illustrative Ex. 3 | (b) `SLIA/name-combine-4-long` | (c) `SLIA/initials-small` |

Figure 5: $KL(Q_{ASAp}\|P)$ and $KL(Q_{GCD}\|P)$

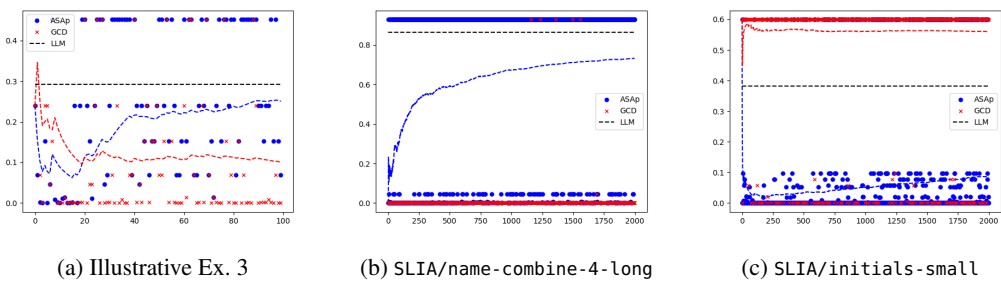

| (a) Illustrative Ex. 3 | (b) `SLIA/name-combine-4-long` | (c) `SLIA/initials-small` |

Figure 6: Expectations of $\tilde{Q}_{ASAp}$, $\tilde{Q}_{GCD}$, and $P$

the latter the grammar is used to restrict constant bit-vectors and operations used to build invariants. Fig. 4 provides examples of SLIA and INV-BV problems. For both families of benchmarks, our prompts consist of 3 in-context examples of the form (specification, solution) and the grammar is then provided as a constraint for GAD. Our third task is the constituency parsing (CP) task already used in prior GCD work [7] where the grammar is used to help the model produce well-parenthesized parse trees for English sentences.

Due to constrained resources and needing to run inference multiple times to measure whether the distribution $\tilde{Q}$ is faithful to $Q$, we randomly select 15 SLIA problems, 15 INV-BV problems, and 6 CP problems. We select the open-source Mistral-7B [12] for evaluation due to its superior reasoning and code generation capabilities.

**Measures.** We run both algorithms for 2,000 iterations/sample on each benchmark.

To assess converge to the target distribution, we measure the Kullback–Leibler (KL) divergence between the distributions of GCD and ASAp from the target distribution $Q$ for a given number of samples. Because the ideal GAD distribution $Q_{P,\mathcal{G}}$ is proportional to the original LLM's distribution $P$ for sequences allowed by a grammar $\mathcal{G}$, we can use the LLM's distribution $P$ on all observed samples as an estimate $Q_{P,\mathcal{G}}$. The quantity $KL(Q\|P)$ only differs by a constant from the KL divergence between empirical distributions and the ideal GAD distribution:

$$KL(\tilde{Q}\|P) = \mathbb{E}_{\tilde{Q}}\left[\log\frac{\tilde{Q}}{P}\right] = \mathbb{E}_{\tilde{Q}}\left[\log\frac{\tilde{Q}}{C\cdot Q_{P,\mathcal{G}}}\right] = \mathbb{E}_{\tilde{Q}}\left[\log\frac{\tilde{Q}}{Q_{P,\mathcal{G}}}\right] - \log C = KL(\tilde{Q}\|Q_{P,\mathcal{G}}) - \log C$$

where $C = \sum_w \mathbb{1}[w \in \mathcal{L}(\mathcal{G})]P(w)$. Thus, $KL(\tilde{Q}\|P)$ can be used to quantify the alignment between the empirical distributions of GCD and ASAp with the ideal GAD distribution.

For example, Fig. 5a shows convergence results for the first 75 iterations on the illustrative Ex. 3—i.e., the KL divergence for $\tilde{Q}_{ASAp}$ quickly converges to 0 whereas the one for $\tilde{Q}_{GCD}$ doesn't.

We also compare the empirical expectations of the variables $\tilde{Q}_{GCD}$, $\tilde{Q}_{ASAp}$, and $P$. For example, Fig. 6a shows convergence results for the first 75 iterations on the illustrative Ex. 3—i.e., $\tilde{Q}_{ASAp}$ converges to the right expectation.

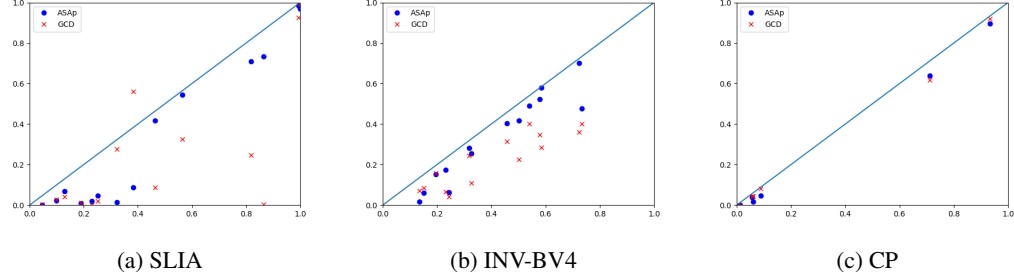

| (a) SLIA | (b) INV-BV4 | (c) CP |

Figure 7: Scatter plots of $\tilde{Q}_{ASAp}$ ($\bullet$) and $\tilde{Q}_{GCD}$ ($\times$) vs. expectations of $P$ after 2,000 samples. Proximity to the diagonal indicates proximity to the actual expectation—e.g., a $\bullet$ at (0.45,0.4) indicates a benchmark where the empirical expectation of $P$ was 0.45 and $\tilde{Q}_{ASAp}$ had converged to an expectation of 0.4 after 2,000 iterations.

**Results.** Fig. 5b and Fig. 6b illustrate a benchmark in which our ASAp algorithm quickly converges to the target distribution. Fig. 5 depicts the KL divergence of a sliding window of size 500 (e.g., the points at x=800 denote the KL divergence of the samples 800-1300). Fig. 6 depicts all the samples from the experiment, as well as how the expectations converges (a point at x=$i$ denotes the empirical expectation on the first $i$ samples. For this case the expecation for GCD stays very close to 0.

Similarly, Fig. 5c and Fig. 6c illustrate a benchmark in which our ASAp algorithm converges slowly. In this case, bot ASAp and GCD are far from the target expectation (Fig. 6c), but because GCD happens to be biased towards the most likely outcome, it exhibits better KL divergence. The complete set of plots is shown in Sec. E.1.

To better understand how the algorithms respectively converge, Fig. 7 plot for each benchmark category the expectations for each benchmark computed by GCD and ASAp against the target expectation of $P$ after 2,000 iterations. The sum of least square difference between expectations computed by GCD and the expectations of $P$ are 2.259 (SLIA), 1.852 (INV-BV4), and 0.109 (CP). The sum of least square difference between expectations computed by ASAp and the expectation and those of $P$ are 1.242 (SLIA), 0.802 (INV-BV4), and 0.159 (CP). While we have too few points for CP to draw conclusions, the expectations computed by ASAp are much closer to the ones computed by GCD across our experiments.

While our work is interested in the theoretical convergence of the ASAp algorithm, we also report how the GCD and ASAp differ for solving the SLIA and INV-BV4 tasks—i.e., how many of the sampled programs are correct solutions to the given problem. GCD and ASAp solve approximately the same set of problems (there is just one SLIA benchmark for which ASAp returns a valid solution on one sample and GCD never does so). ASAp produces correct samples 38% more often than GCD (geomean), whereas for SLIA benchmarks that both tools can solve, ASAp produces correct samples 73% less often than GCD (geomean). Detailed results can be found in Sec. E.2. These results are in line with the fact ASAp shows faster convergence on INV-BV4 benchmarks. For example, for the benchmark illustrated in Fig. 5b, ASAp returns the correct solution for 1588 samples, whereas GCD only returns the correct solution 12 times, whereas for the benchmark in Fig. 5c, ASAp returns the correct solution 69 times and GCD 363 times.

**Discussion and Limitations.** As predicted by our theorems, on most benchmarks the ASAp algorithm converges to the desired distribution $P$ whereas GCD does not improve over time (i.e., it exhibits the bias described in this paper).

While ASAp has no strong effect on solving downstream tasks, we observe that on instances where the convergence is prominent, ASAp ends up sampling correct solutions more often than GCD, which is what we expect when the LLM has "learned" how to solve the given task.

The key limitation of our work is the current slow convergence of the ASAp algorithm. In some benchmarks, even after 2,000 iterations the KL divergence barely improves and even though the expectation of $\tilde{Q}_{ASAp}$ is improving, it converges very slowly.

We highlight that the contributions of this paper are discovering and formalizing the bias of existing constrained decoding approaches and proposing the first converging algorithm to address this problem.

Now that we have identified the problem, there are many "low-hanging fruits" to improve our sampling strategy, which are great targets for future work—e.g., using forms of targeted beam search to bootstrap our sample set to better explore grammar paths and avoid sampling similar strings.

## 5    Related Work

**Constrained Decoding**    Past work has extensively explored *constrained decoding* algorithms, which modify the original decoding process of LLMs to ensure the output adheres to a user-specified regular [18, 28] or context-free language [5, 6, 7, 19, 23, 24, 25, 26] in a discrete space. Other works enforce hard output constraints using dynamic monitoring and verification methods [1, 15, 27] or by modifying beam search techniques to impose lexical constraints, which require specific keywords to appear in the generated text [4, 9, 10, 16, 17, 20]. At a high level, these methods involve running the LLM decode in parallel with a monitoring scheme (e.g., parsing algorithms for CFGs) to identify which next tokens or beams can produce valid output sequences that meet the constraints. The decoder then masks out any tokens that would lead to invalid sequences, sampling only from the permissible ones.

To incorporate sequence-level soft semantic or contextual constraints, Amini et al. [3], Kumar et al. [13], Li et al. [14], Qin et al. [21] have applied gradient-based sampling techniques that relax those constraints to differentiable ones, used them as classifiers to further guide the decoding process. While these works guarantee that the decoded output meets the specified constraints (whether in the form of grammar, monitoring schemes, or differentiable functions), they often operate greedily and introduce bias into the output distribution in the way that has been discussed in this paper. Depending on the application one considers, this problem may or may not affect downstream tasks, but as we have argued in this paper, the bias can be quite prominent and sometimes affect downstream performance. Our adaptive decoding algorithm improves decoding over time by analyzing how previous samples led to nongrammaticaility.

**Constraint-Aligned Decoding**    This paper formally defines the problem of aligning the output distribution of an LLM in the presence of a constraint. We focus our attention on constraints expressed as grammars, but our definitions and algorithm apply to any constraint for which possible satisfaction (in our case grammaticality) can be evaluated in a left-to-right manner.

In some settings, one is interested in generating multiple outcomes with an LLM to approximate a distribution of interest [11, 22]—e.g., to generate a random number or a set of good test cases for a program. As we have shown, constrained decoding can heavily skew the LLMs distribution and result in biasing the model towards certain constraint-matching sequences. While our work is at this point theoretical, now that the problem of aligning an LLM's distribution with constraints has been defined, we expect advances in how sampling is performed to quickly converge to better distributions faster (e.g., using beam search to quickly explore possible paths instead of just sampling).

## 6    Conclusion

We have introduced a new analysis of the ideal target for constrained sampling from an LLM using a grammar, which we call grammar-aligned decoding (GAD). We proposed a new algorithm for GAD which we call ASAp that iteratively builds better approximations to the critical re-weighting term required for GAD: the expected future grammaticality. We analyzed the convergence of our proposed algorithm and demonstrated its effectiveness in relation to existing grammar-constrained decoding techniques on a set of benchmark code generation tasks. We analyzed and evaluated our approach using constraints enforced by a context-free grammar; however, extensions of our approach might be applied to more general classes of constraints for LLM decoding.

While the primary goals of this work are to formalize the likelihood misalignment problem of existing grammar-constrained decoding approaches and to provide an initial solution with provable asymptotic guarantees, future work may explore faster-converging approaches, such as sampling multiple tokens simultaneously, to improve efficiency further. We hope this work lays a solid foundation for generating structured outputs from LLMs without distorting the original distribution, advancing the field toward more efficient, trustworthy, and constraint-aligned approaches in LLM-driven generation.

**Acknowledgement**

The authors would like to thank NeurIPS anonymous reviewers for their insightful feedback and helpful discussions. The authors thank Gurindar S. Sohi, Shivaram Venkataraman, Ming Liu, and Yixuan Li for the support of computing resources. This work is supported, in part, by NSF under grants CCF-1918211, CCF-1955457, CCF-2023222, CCF-2200333, CCF-2211968, CCF-2402833, CCF-2422214, and CCF-2446711. Any opinions, findings, and conclusions or recommendations expressed in this publication are those of the authors, and do not necessarily reflect the views of the sponsoring entities.

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

# Appendix

## A  Hardware and Software

Our experiments are conducted on 4 NVIDIA RTX A6000 GPUs and 4 NVIDIA A100 GPUs. Our implementation is based on Python 3.10 and PyTorch 2.1.2.

## B  Hyperparameters

The hyperparameters discussed in this paper pertain to the decoding strategy of language models. As we aim to investigate the LM's original distribution, we set Top-P at 1.0, Temperature at 1.0, and Top-K at 0 to consider the complete token vocabulary.

## C  Model Checkpoint

We use the Mistral-7B model checkpoint provided by Hugging Face: `https://huggingface.co/mistralai/Mistral-7B-Instruct-v0.2`.

## D  Experimental Details

### D.1  SLIA and INV-BV

**Prompt Construction**  For both families of benchmarks, our prompts adopt standard in-context learning format which consist of 3 in-context examples of the form (specification, solution) and ask the model to provide the solution for the last example. A concrete example would be

```
You are an expert in program synthesis.
You are tasked with solving a Syntax-Guided Synthesis (SyGuS) problem.
Your goal is to output a function that should produce outputs that satisfy
a series of constraints when given specific inputs.

Question:
(set-logic BV)

(synth-fun inv ((s (BitVec 4)) (t (BitVec 4))) (BitVec 4))

(declare-var s (BitVec 4))
(declare-var t (BitVec 4))
(define-fun udivtotal ((a (BitVec 4)) (b (BitVec 4))) (BitVec 4)
    (ite (= b #x0) #xF (bvudiv a b)))
(define-fun uremtotal ((a (BitVec 4)) (b (BitVec 4))) (BitVec 4)
    (ite (= b #x0) a (bvurem a b)))
(define-fun min () (BitVec 4)
    (bvnot (bvlshr (bvnot #x0) #x1)))
(define-fun max () (BitVec 4)
    (bvnot min))
(define-fun l ((s (BitVec 4)) (t (BitVec 4))) Bool
    (bvsle (bvlshr s (inv s t)) t))
(define-fun SC ((s (BitVec 4)) (t (BitVec 4))) Bool
    (or (bvult t min) (bvsge t s)))
(constraint (=> (SC s t) (l s t)))

(check-synth)
Solution:
(define-fun inv ((s (BitVec 4)) (t (BitVec 4))) (BitVec 4) (bvnot (bvor s #b0111)))

... (2 more examples)
```

```
Question:
(set-logic BV)

(synth-fun inv ((s (BitVec 4)) (t (BitVec 4))) (BitVec 4))

(declare-var s (BitVec 4))
(declare-var t (BitVec 4))
(define-fun udivtotal ((a (BitVec 4)) (b (BitVec 4))) (BitVec 4)
    (ite (= b #x0) #xF (bvudiv a b)))
(define-fun uremtotal ((a (BitVec 4)) (b (BitVec 4))) (BitVec 4)
    (ite (= b #x0) a (bvurem a b)))
(define-fun min () (BitVec 4)
    (bvnot (bvlshr (bvnot #x0) #x1)))
(define-fun max () (BitVec 4)
    (bvnot min))
(define-fun l ((s (BitVec 4)) (t (BitVec 4))) Bool
    (bvsgt (bvnot (inv s t)) t))
(define-fun SC ((s (BitVec 4)) (t (BitVec 4))) Bool
    (distinct t max))
(constraint (=> (SC s t) (l s t)))

(check-synth)
Solution:
```

**Grammar Constraint**   While most SYGUS problems contain grammar constraints, some problems
have grammars implicitly defined by the theory. We explicitly converted the grammar constraint of
the problem into EBNF format for constrained-decoding. The example for the last example would be

```
root ::= "(define-fun inv ((s (BitVec 4)) (t (BitVec 4))) (BitVec 4) " Start ")"
Start ::= "s" | "t" | "#x0" | "#x8" | "#x7"
        | "(" "bvneg" " " Start ")" | "(" "bvnot" " " Start ")"
        | "(" "bvadd" " " Start " " Start ")" | "(" "bvsub" " " Start " " Start ")"
        | "(" "bvand" " " Start " " Start ")" | "(" "bvlshr" " " Start " " Start ")"
        | "(" "bvor" " " Start " " Start ")" | "(" "bvshl" " " Start " " Start ")"
```

## D.2   Constituency Parsing

For Constituency parsing task, our prompts consist of 8 in-context examples of the form. A concrete
example would be

```
Perform constituency parsing on the provided sentences in accordance with the Penn TreeBank
annotation guidelines. Fill in the last mapping.

Ad Notes
->
[ ( NP-HLN ( NN Ad ) ( NNS Notes ) ) ]

The market crumbled
->
[ ( S ( NP-SBJ ( DT The ) ( NN market ) ) ( VP ( VBD crumbled ) ) ) ]

I felt betrayed he later said
->
[ ( S ( S-TPC-1 ( NP-SBJ ( PRP I ) ) ( VP ( VBD felt ) ( ADJP-PRD ( VBN betrayed ) ) ) )
( NP-SBJ ( PRP he ) ) ( ADVP-TMP ( RB later ) ) ( VP ( VBD said ) ) ) ]

Friday October 13 1989
->
[ ( NP ( NNP Friday ) ( NNP October ) ( CD 13 ) ( CD 1989 ) ) ]
```

```
The Arabs had merely oil
->
[ ( S ( NP-SBJ ( DT The ) ( NNPS Arabs ) ) ( VP ( VBD had )
( NP ( RB merely ) ( NN oil ) ) ) ) ]

Energy
->
[ ( NP-HLN ( NN Energy ) ) ]

Some U.S. entrepreneurs operate on a smaller scale
->
[ ( S ( NP-SBJ ( DT Some ) ( NNP U.S. ) ( NNS entrepreneurs ) ) ( VP ( VBP operate )
( PP-MNR ( IN on ) ( NP ( DT a ) ( JJR smaller ) ( NN scale ) ) ) ) ) ]

Knowledgeware Inc.
->
[ ( NP-HLN ( NNP Knowledgeware ) ( NNP Inc. ) ) ]

They are more sophisticated this time
->
```

**Grammar Constraint**   For the constituency parsing (CP) task we used the grammar provided in prior GCD work [7]. The grammar is too large to attach, but it is used to help the model produce well-parenthesized parse trees and ensure that all words in a given English sentence appear in left-to-right order.

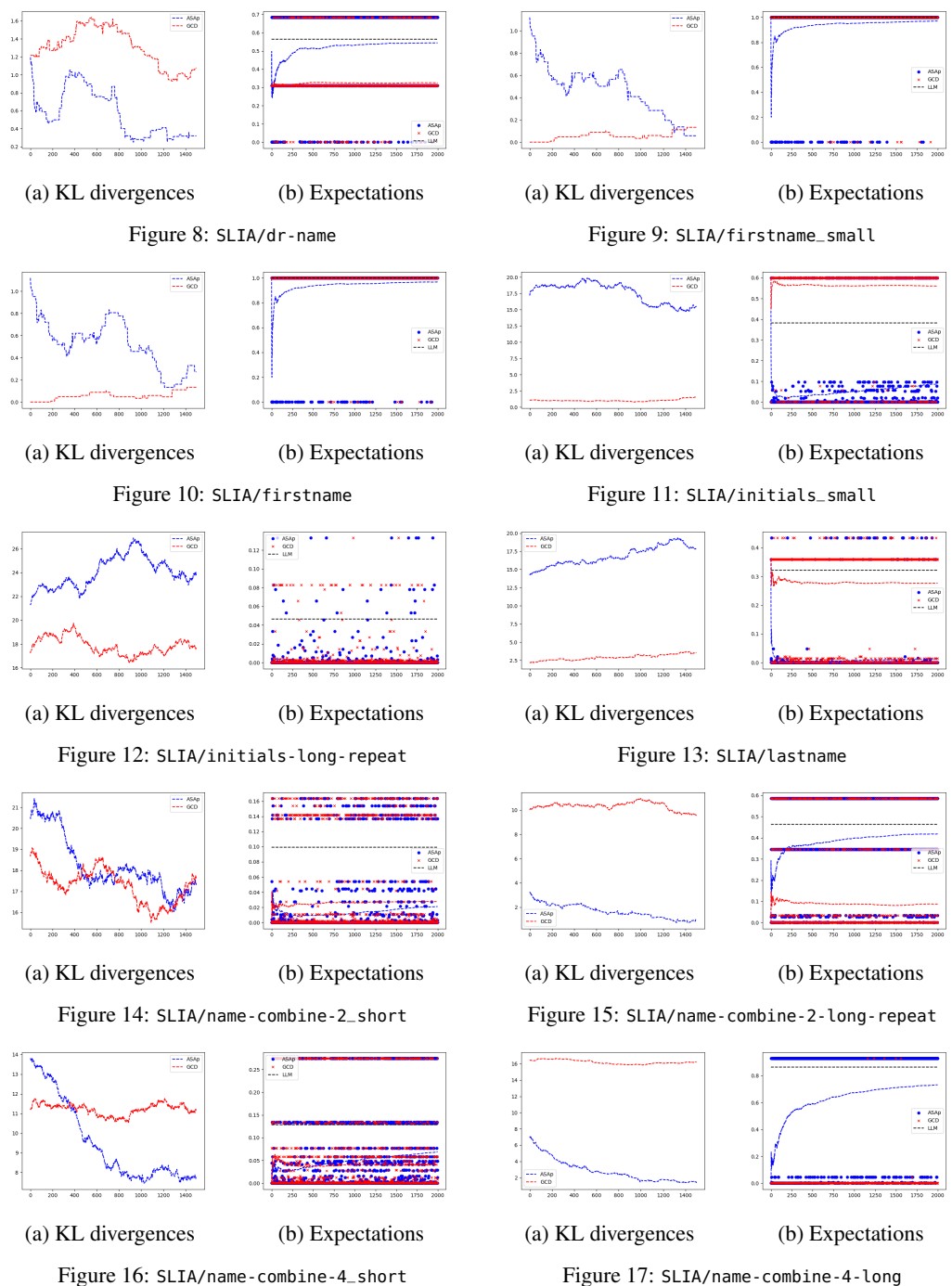

(a) KL divergences     (b) Expectations

Figure 8: `SLIA/dr-name`

(a) KL divergences     (b) Expectations

Figure 9: `SLIA/firstname_small`

(a) KL divergences     (b) Expectations

Figure 10: `SLIA/firstname`

(a) KL divergences     (b) Expectations

Figure 11: `SLIA/initials_small`

(a) KL divergences     (b) Expectations

Figure 12: `SLIA/initials-long-repeat`

(a) KL divergences     (b) Expectations

Figure 13: `SLIA/lastname`

(a) KL divergences     (b) Expectations

Figure 14: `SLIA/name-combine-2_short`

(a) KL divergences     (b) Expectations

Figure 15: `SLIA/name-combine-2-long-repeat`

(a) KL divergences     (b) Expectations

Figure 16: `SLIA/name-combine-4_short`

(a) KL divergences     (b) Expectations

Figure 17: `SLIA/name-combine-4-long`

# E   Detailed Experimental Results

We provide additional plots and experimental data.

## E.1   Plots

Figures 8–22 provide the KL divergence and expectation results for the SLIA benchmarks. Figures 23–37 provide the KL divergence and expectation results for the INV-BV benchmarks. Figures 38–43 provide the KL divergence and expectation results for the INV-BV benchmarks.

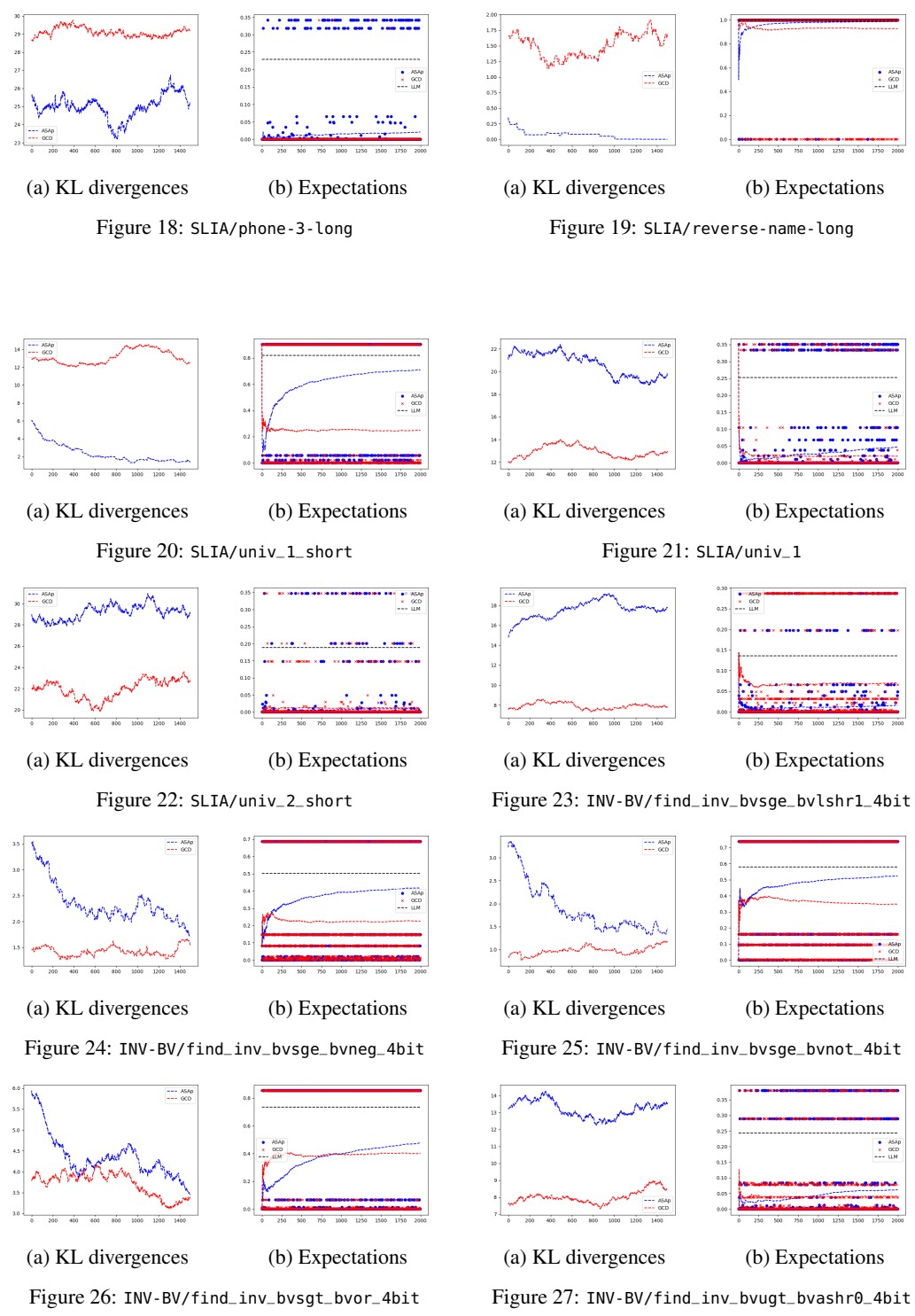

(a) KL divergences      (b) Expectations

Figure 18: `SLIA/phone-3-long`

(a) KL divergences      (b) Expectations

Figure 19: `SLIA/reverse-name-long`

(a) KL divergences      (b) Expectations

Figure 20: `SLIA/univ_1_short`

(a) KL divergences      (b) Expectations

Figure 21: `SLIA/univ_1`

(a) KL divergences      (b) Expectations

Figure 22: `SLIA/univ_2_short`

(a) KL divergences      (b) Expectations

Figure 23: `INV-BV/find_inv_bvsge_bvlshr1_4bit`

(a) KL divergences      (b) Expectations

Figure 24: `INV-BV/find_inv_bvsge_bvneg_4bit`

(a) KL divergences      (b) Expectations

Figure 25: `INV-BV/find_inv_bvsge_bvnot_4bit`

(a) KL divergences      (b) Expectations

Figure 26: `INV-BV/find_inv_bvsgt_bvor_4bit`

(a) KL divergences      (b) Expectations

Figure 27: `INV-BV/find_inv_bvugt_bvashr0_4bit`

## E.2 Correctness Results for SYGUS Tasks

Table 1 shows how many samples (out of 2000) yielded correct solutions for each benchmark (bold is better). The task initials_long-repeat was only solved using ASAp.

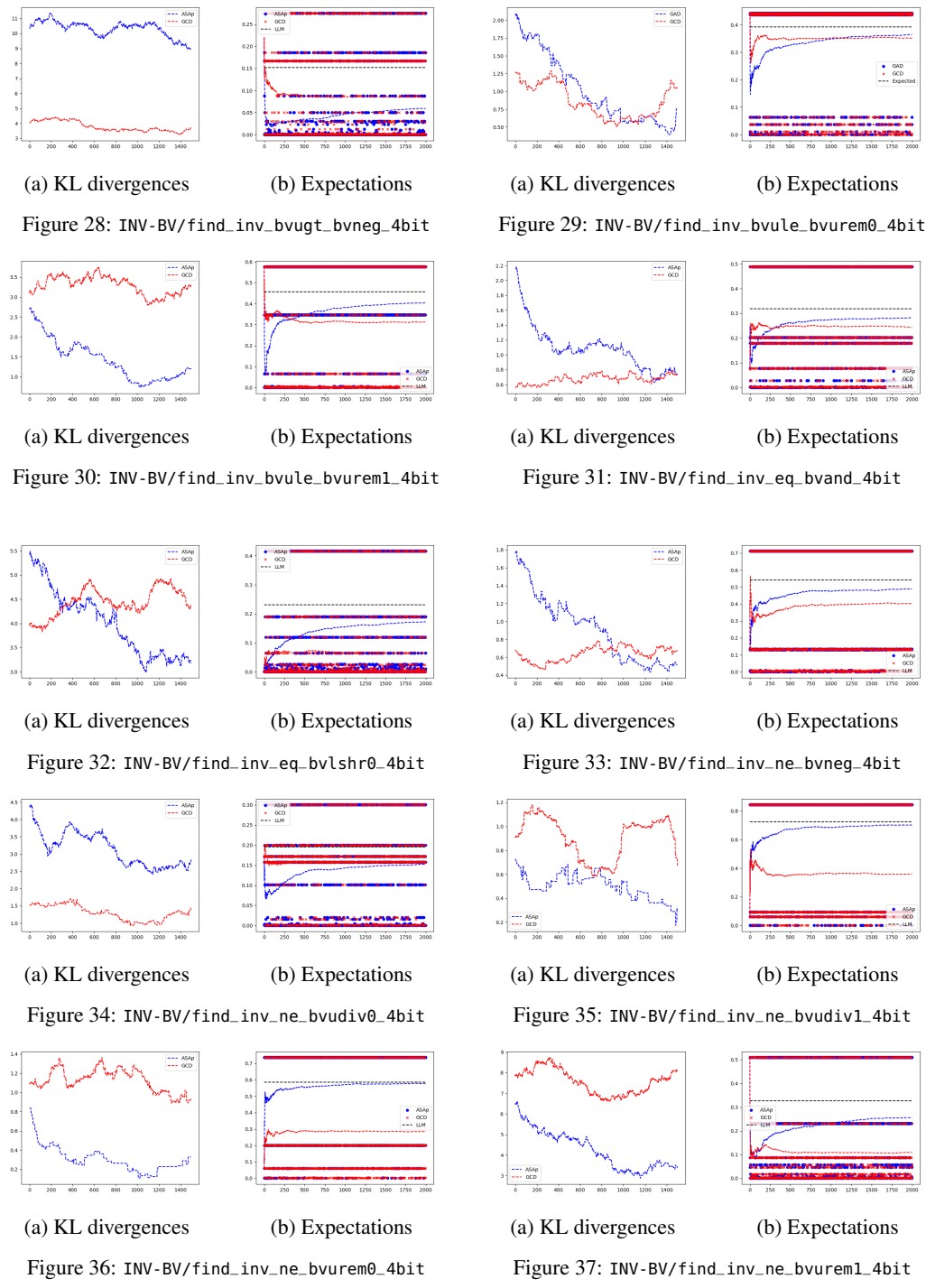

(a) KL divergences    (b) Expectations

Figure 28: `INV-BV/find_inv_bvugt_bvneg_4bit`

(a) KL divergences    (b) Expectations

Figure 29: `INV-BV/find_inv_bvule_bvurem0_4bit`

(a) KL divergences    (b) Expectations

Figure 30: `INV-BV/find_inv_bvule_bvurem1_4bit`

(a) KL divergences    (b) Expectations

Figure 31: `INV-BV/find_inv_eq_bvand_4bit`

(a) KL divergences    (b) Expectations

Figure 32: `INV-BV/find_inv_eq_bvlshr0_4bit`

(a) KL divergences    (b) Expectations

Figure 33: `INV-BV/find_inv_ne_bvneg_4bit`

(a) KL divergences    (b) Expectations

Figure 34: `INV-BV/find_inv_ne_bvudiv0_4bit`

(a) KL divergences    (b) Expectations

Figure 35: `INV-BV/find_inv_ne_bvudiv1_4bit`

(a) KL divergences    (b) Expectations

Figure 36: `INV-BV/find_inv_ne_bvurem0_4bit`

(a) KL divergences    (b) Expectations

Figure 37: `INV-BV/find_inv_ne_bvurem1_4bit`

# F    Will ASAp still be more aligned than GCD after fine-tuning?

## F.1    Experimental setup for fine-tuning

We adhere to the established QLoRA finetuning pipeline and create task-specific datasets of INV-BV4 and CP for instruction tuning. In line with our paper's methodology, we incorporate in-context examples in the instruction tuning dataset to enhance the models' performance in in-context learning. For each task, we independently finetune Mistral-7B, resulting in two versions of the model (for

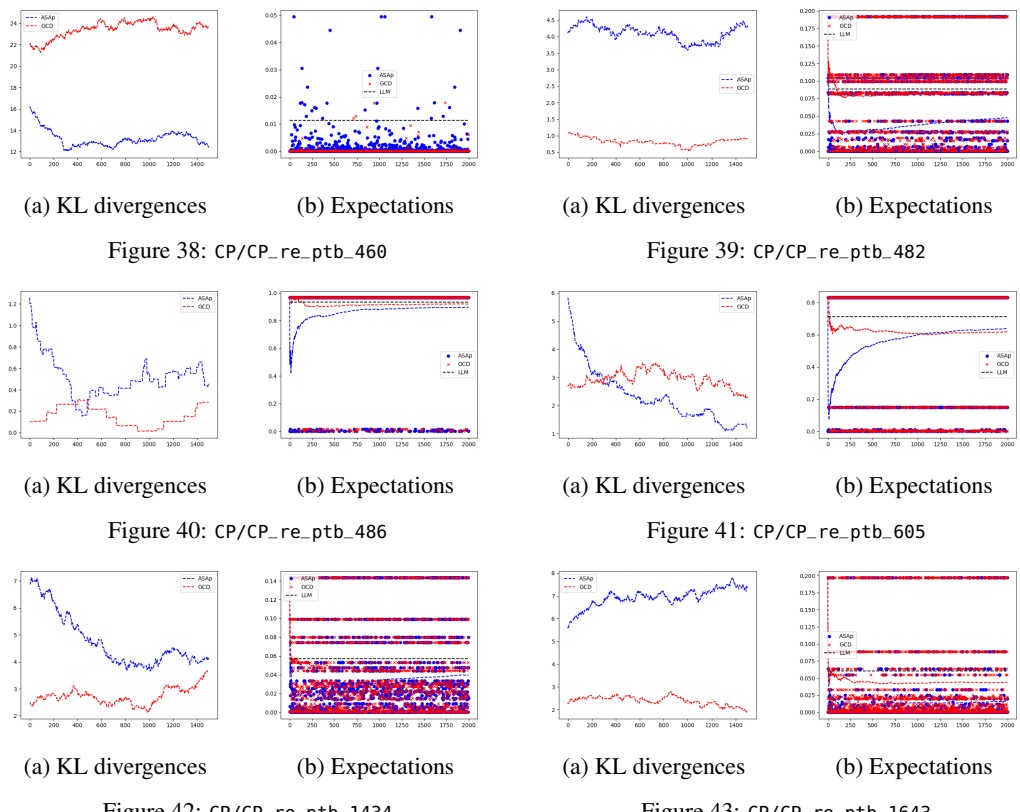

(a) KL divergences  (b) Expectations

Figure 38: CP/CP_re_ptb_460

(a) KL divergences  (b) Expectations

Figure 39: CP/CP_re_ptb_482

(a) KL divergences  (b) Expectations

Figure 40: CP/CP_re_ptb_486

(a) KL divergences  (b) Expectations

Figure 41: CP/CP_re_ptb_605

(a) KL divergences  (b) Expectations

Figure 42: CP/CP_re_ptb_1434

(a) KL divergences  (b) Expectations

Figure 43: CP/CP_re_ptb_1643

INV-BV4 and CP). We employ a standard train-validation-test split of 70-10-20%. Instruction tuning is conducted on the training set, and model selection is based on the lowest validation loss. Key hyperparameters include a learning rate of 2e-4, a warmup ratio of 0.03, a maximum sequence length of 2048, LoRA alpha of 32, LoRA dropout of 0.05, and LoRA rank of 64. The best checkpoints were at 328 and 536 steps for INV-BV and CP, respectively.

## F.2    Additional Results

**No significant differences in convergence rates post fine-tuning.**    In Section 4, we evaluate ASAp and GCD on the base model Mistral-7B. A natural extension of this evaluation is determining whether ASAp retains its advantages over GCD after fine-tuning the base model on task-specific datasets, which optimizes the LLM for higher grammatical accuracy from the start.

In our fine-tuning step, we want to teach the LLM to assign higher probabilities to grammatical outputs for the specific task DSL. We randomly selected two INV-BV problems (find_inv_bvsge_bvneg_4bit and find_inv_bvsgt_bvor_4bit for INV-BV) and four CP problems (CP_re_ptb_215, CP_re_ptb_434, CP_re_ptb_1627 and CP_re_ptb_1643 for CP) from the test set, and instrction tuned input-output pairs of prompt and output programs in the training set for the base model Mistral-7B. We obtained two fine-tuned models, one for INV-BV and one for CP.

We tested GCD and ASAp on the finetuned Mistral-7B on the randomly left-out problems and checked the convergence rates of the KL-divergence. The results from finetuned Mistral-7B did not show significant differences in terms of convergence compared to the base Mistral-7B. As done in Section 4, we computed the expectation for each benchmark obtained via GCD and ASAp after 2,000 iterations and compared it against the target expectation $Q^{P,\mathcal{G}}$ of GAD. The sum of least squares difference between expectations computed by GCD and the expectations of $Q^{P,\mathcal{G}}$ are 0.677 (INV-BV4), 0.278 (CP), while ASAp achieved lower errors: 0.051 (INV-BV4), 0.201 (CP), indicating that ASAp more closely aligned with the exact GAD expectations. We did not include SLIA as we did not have sufficient data for further fine-tuning.

Table 1: Correctness of solutions for different algorithms.

| | Benchmark | Correct ASAP | Correct GCD |
|---|---|---|---|
| SLIA | phone-3-long | 0 | 0 |
| | name-combine-2_short | 171 | **319** |
| | name-combine-2-long-repeat | 0 | 0 |
| | name-combine-4-short | **20** | 11 |
| | name-combine-4-long | **1588** | 12 |
| | lastname | 285 | **1526** |
| | firstname | 1960 | **1997** |
| | firstname_small | 1754 | **1997** |
| | reverse-name-long | **1981** | 1859 |
| | univ_1 | **67** | 40 |
| | univ_1_short | 605 | **1859** |
| | univ_2_short | 0 | 0 |
| | dr-name | 357 | **1654** |
| | initials_small | 69 | **363** |
| | initials_long | 540 | **1584** |
| | initials_long-repeat | **3** | 0 |
| INV-BV | find_inv_ne_bvudiv1_4bit | 0 | 0 |
| | find_inv_bvugt_bvashr0_4bit | **83** | 49 |
| | find_inv_eq_bvlshr0_4bit | **635** | 228 |
| | find_inv_eq_bvand_4bit | **1599** | 1305 |
| | find_inv_bvule_bvurem0_4bit | **1813** | 1710 |
| | find_inv_bvsgt_bvor_4bit | **11** | 10 |
| | find_inv_bvugt_bvneg_4bit | **84** | 36 |
| | find_inv_bvule_bvurem1_4bit | 143 | **227** |

