# OpenReview forum: "Grammar-Aligned Decoding"
_NeurIPS.cc/2024/Conference — NeurIPS 2024 poster_

### Official Review · Reviewer_gp9U · 2024-07-07

**Soundness:** 2
**Presentation:** 3
**Contribution:** 3
**Rating:** 5
**Confidence:** 4

**Summary:**

This paper focuses on the constrained decoding scenario where LLMs are expected to produce high-quality and grammatically correct outputs. It presents the adaptive sampling with approximate expected futures (ASAp) method, which is designed to enhance the quality of the output by adjusting the conditional probability to align with the model’s original output distribution while ensuring grammaticality. Central to ASAp is the development of effective approximations to the expected future grammaticality (EFG). More specifically, ASAp involves an extra sampling process, where the estimates of EFG are iteratively refined after each sampling and will eventually converge to exact EFG in the limit of infinite iterations. Experimental results show that the output probability distribution of ASAp is closer to the original one than that of GCD methods.

**Strengths:**

Advantages of ASAp include facilitating more accurate grammatical predictions. ASAp approximates the desired distribution by considering the potential future grammaticality of sequences, whereas GCD typically samples greedily based on current probabilities without considering the future development of the sequence. This approach allows ASAp to more accurately predict and generate text sequences that comply with grammatical rules.

**Weaknesses:**

The algorithm heavily relies on prior samples to estimate grammatical probabilities, which can be computationally expensive and difficult to manage, especially in cases where the dataset is large or the environment is dynamically changing. Moreover, a fair comparison of decoding speed between ASAp and GCD methods is required, as ASAp needs this sampling process to produce high-quality outputs, but conducting sampling on LLMs is very time-consuming.

Only evaluating the quality of the output by KL divergence and expectations is limited because the decoding sampling algorithm, except for the greedy search, does not always obey the probability distribution. For example, when the decoding temperature is set to 1, models may not output tokens with the highest probability. To conduct a more intuitive evaluation, one can choose some constrained decoding tasks with explicit quality scores, such as constrained translation [1] with BLEU and Exact Match, and code generation, which can be evaluated by the passing rate.

Since the experiments in this paper are based on prompt-motivated LLMs, a natural step is to fine-tune LLMs and see if they can produce grammatically correct output with high quality. On the other hand, the authors can also try ASAp based on fine-tuned LLMs, which may further boost the performance of ASAp.

To enhance the readability of the article, the following suggestions can be considered.
1. It’s more intuitive to give specific input and output examples for SLIA and INV-BV tasks.
2. The names GCD and GAD are too similar to easily differentiate, even after reading the paper.
3. Other typos: In line 289, the word “both” is typed as “bot”.

[1] Wang, S., Li, P., Tan, Z., Tu, Z., Sun, M., & Liu, Y. (2022). A Template-based Method for Constrained Neural Machine Translation. arXiv preprint arXiv:2205.11255.

**Questions:**

Given previous samples with the prefix $w_{1:i}$, there exist abundant future tokens beyond just the next token. However, ASAp only uses the next token likelihoods to facilitate a better estimate of EFG. Why not use other future tokens?

**Limitations:**

Yes

---

> ### Author Rebuttal · Authors · 2024-08-07
>
> > The algorithm heavily relies on prior samples to estimate grammatical probabilities, which can be computationally expensive and difficult to manage… a fair comparison of decoding speed between ASAp and GCD methods is required, as…conducting sampling on LLMs is very time-consuming.
>
> Indeed, ASAp requires building a large data structure to keep track of the sampled prefixes, but decoding **each sample takes the same time with GCD and ASAp**.
>
> > Only evaluating the quality of the output by KL divergence and expectations is limited because the decoding sampling algorithm, except for the greedy search, does not always obey the probability distribution.. one can choose some constrained decoding tasks with explicit quality scores..
>
> We agree that downstream, extrinsic evaluations are of interest: e.g., how much does ASAp improve task performance metrics on specific tasks? (See next paragraph, we do include some of this information in our paper.) However, these results are task dependent, as the reviewer mentions -- in some cases, increasing likelihood may correspond to higher task accuracy and in some cases not.
> Since we focus on formalizing the problem of distribution misalignment in GCD, the primary objective of our experiments has been to evaluate **intrinsic** measures, i.e., to assess how closely the probability distributions from GCD and ASAp approximate the exact GAD in Equation 2 in practice.
>
> We do present the downstream quality of the output in Table 1 in Appendix A.5.2. However, since the model we used for evaluation was not sufficiently well-trained for our task, the experimental results in Appendix A.5 show that the output with the highest probability does not necessarily indicate good quality on the downstream task.
>
> > a natural step is to fine-tune LLMs and see if they can produce grammatically correct output with high quality. On the other hand, the authors can also try ASAp based on fine-tuned LLMs
>
> We thank the reviewer’s suggestions regarding the evaluation. Previous work on GCD has shown that grammar-constrained LMs outperform fine-tuned models in structured NLP tasks (citation [5]). But we agree it is interesting to find out whether ASAp still provides benefits over GCD when applied to an LLM distribution that has **already been fine-tuned** to achieve higher grammaticality from the outset.
>
> Due to limited time and data (139 INV-BV and 2416 CP problems), we conducted a small experiment to test the reviewer’s hypothesis. In our finetuning step, we want to teach the LLM to assign higher probabilities to grammatical outputs for the specific task DSL. We randomly selected 2 INV-BV problems (find_inv_bvsge_bvneg_4bit and find_inv_bvsgt_bvor_4bit for INV-BV) and 4 CP problems (CP_re_ptb_215, CP_re_ptb_434, CP_re_ptb_1627 and CP_re_ptb_1643 for CP) from the test set, and used all other input-output pairs of prompt and output programs to construct datasets for finetuning the base LLM.  We obtained 2 finetuned LLMs, one for INV-BV and one for CP.
>
> We ran GCD and ASAp on the finetuned models on the randomly left-out problems and checked the convergence rates of the KL-divergence. The results from finetuned models did not show significant differences in terms of convergence compared to the original model. As done in our evaluation, we computed the expectation for each benchmark obtained via GCD and ASAp after 2,000 iterations and compared it against the target expectation $Q^{P,G}$ (line 76) of GAD. The sum of least squares difference between expectations computed by GCD and the expectations of $Q^{P,G}$ are 0.677 (INV-BV4), 0.278 (CP) (respectively. 0.051 (INV-BV4), 0.201 (CP) for ASAp). I.e., the expectations computed by ASAp were closer to the expectations of exact GAD than those computed by GCD. We didn’t include SLIA as we didn’t have sufficient data for further finetuning.
>
> We acknowledge there are alternative ways to fine-tune the model for learning grammatically; this goal is beyond the scope of the paper.
>
> **Details on experimental setup**: We adhere to the established LoRA finetuning pipeline and create task-specific datasets for instruction tuning. In line with our paper's methodology, we incorporate in-context examples in the instruction tuning dataset to enhance the models' performance in in-context learning. For each task, we independently finetune Mistral-7B, resulting in two versions of the model (for INV-BV4 and CP). We employ a standard train-validation-test split of 70-10-20%. Instruction tuning is conducted on the training set, and model selection is based on the lowest validation loss. Learning rate: 2e-4, warmup ratio: 0.03, max sequence length: 2048, LoRA alpha: 32, LoRA dropout: 0.05, and LoRA r: 64. The best checkpoints for the finetuned models for INV-BV and CP are at 328 and 536 steps, respectively.
>
> > It’s more intuitive to give specific input and output examples for SLIA and INV-BV tasks.
>
> Due to the space constraint in the main paper, we have included concrete examples and the grammar used to constrain decoding for LLMs in Appendices A.4.1 and A.4.2. We will provide a detailed description of these problems in the supplementary materials.
>
> > The names GCD and GAD are too similar to easily differentiate, even after reading the paper.
>
> We will use more readable macros.
>
> > Given previous samples with the prefix w_{1:i}, there exist abundant future tokens beyond just the next token. However, ASAp only uses the next token likelihoods to facilitate a better estimate of EFG. Why not use other future tokens?
>
> The reviewer is right that there could be faster-converging decoding approaches, which we are exploring as future work (e.g., as hinted by the reviewer, sampling more tokens at once to accelerate convergence is one of them).
>
> The main focus of the paper is to formalize the likelihood misalignment problem in existing grammar-constrained decoding, and to provide an initial solution with provable asymptotic guarantees.

---

> > ### Comment · Reviewer_gp9U · 2024-08-12
> >
> > Thank you for your response, which has addressed some of my concerns. I think this is an interesting work. However, I will maintain my rating score as the issues warrant consideration in a major revision of this submission.

---

### Official Review · Reviewer_CJH3 · 2024-07-12

**Soundness:** 4
**Presentation:** 4
**Contribution:** 3
**Rating:** 7
**Confidence:** 3

**Summary:**

This paper proposes adaptive sampling with approximate expected futures (ASAp) for grammar-aligned decoding for LLMs. The main objective of this method is to match the conditional probability of the LLM’s distribution conditioned on the given grammar. The evaluation is performed on code generation and structured NLP tasks showing a better likelihood for LLM’s outputs being grammar-constrained.

**Strengths:**

- The method is well-formalized and clear. The examples are very useful for faster understanding
- The method is well-motivated (there isn’t an independent motivation section but each choice in the design of the method is appropriately justified)
- The paper provides empirical validation that the method works and that it improves the benchmark scores

**Weaknesses:**

- Nothing in particular. Maybe I have found Section 2 of the paper somewhat difficult to read but it is probably because of my lack of recent readings on CFG.

**Questions:**

I suggest maybe some improvement in Section 2. Restructuring it maybe and providing slightly more context on CFG (or giving a concrete example of how it works as a reminder, I found Fig 1 not illustrative enough), I'm sure I won't be the only reader who doesn't have recent experience with CFG.

**Limitations:**

Yes, limitations are discussed. What I liked in particular in this paper is that the limitations of ASAp are well exposed and discussed throughout the paper and not just in the limitations section.

---

> ### Author Rebuttal · Authors · 2024-08-07
>
> > I suggest maybe some improvement in Section 2. Restructuring it maybe and providing slightly more context on CFG (or giving a concrete example of how it works as a reminder, I found Fig 1 not illustrative enough), I'm sure I won't be the only reader who doesn't have recent experience with CFG.
>
> We thank the reviewer for the suggested improvement. In the revision, we will include a formal definition of CFG and an example of how a string can be derived for the CFG in Fig 1.

---

### Official Review · Reviewer_cHBv · 2024-07-12

**Soundness:** 4
**Presentation:** 3
**Contribution:** 3
**Rating:** 6
**Confidence:** 4

**Summary:**

This paper points out that prior methods on grammar-constrained decoding (GCD) distorts the language model's learned distribution over sequences. At the heart of this exposition is the notion of _expected future grammaticality_ (EFG), where prior GCD methods can be cast an an upper-bound approximation to the EFG. To ameliorate this problem, the authors proposed the ASAp method: through many iterations, the EFG of a prefix can be better estimated.

**Strengths:**

This is an important finding that conventional methods on grammar-constrained decoding distorts the language model's learned distribution. As far as I know, this important fact has not been pointed out before. The notion of expected future grammatically will be impactful in the area of sequence decoding.

**Weaknesses:**

The proposed method seems very slow to run, requiring many iterations. Additionally, it requires storing a table of all seen prefixes and their future grammatically, which would be very large if the grammar itself is large.

It seems to me that the ASAp method only depends on a grammar -- it does not specifically requires a training or test set to run. I wonder if it would be possible to first sample extensively and train the ASAp EFG estimates, then decode on a set. In this way ASAp can be cast as a preprocessing step of the grammar: if it is very slow, it wouldn't matter much since it'll only be processed once.

For example we are doing NL2SQL under many conditions. The target grammar stays the same: SQL. One can utilize a LM and generate massive amount of prefixes that has EFG > 0, then compute this estimation of $\tilde c_S$ for all these distinct prefixes.

**Questions:**

- L10: "prorblem" => "problem"
 - Missing related work: R Shin, et al (2021): Constrained language models yield few-shot semantic parsers. https://aclanthology.org/2021.emnlp-main.608/
 - Alg 1: What is "ancestral sampling"? Please explain.

**Limitations:**

The authors adequately discussed the limitations of their work.

---

> ### Author Rebuttal · Authors · 2024-08-07
>
> > The proposed method seems very slow to run, requiring many iterations. Additionally, it requires storing a table of all seen prefixes and their future grammatically, which would be very large if the grammar itself is large. It seems to me that the ASAp method only depends on a grammar -- it does not specifically require a training or test set to run. I wonder if it would be possible to first sample extensively and train the ASAp EFG estimates, then decode on a set.
>
> We acknowledge that our proposed method can be slow to converge. However, the main focus of the paper is to formalize the likelihood misalignment problem in existing grammar-constrained decoding, and to provide an initial solution to address this problem together with a proof of convergence. The reviewer is right that there could be better decoding approaches, and we have started exploring several approaches as part of our future work (e.g., as hinted by the reviewer, sampling more heterogeneous sequences during preprocessing, and training a neural estimate of expected future grammaticality, offline, to use later during test-time sampling).
>
> > Missing related work: R Shin, et al (2021): Constrained language models yield few-shot semantic parsers.
>
> We thank the reviewer for providing the missing early related work on grammar-constrained decoding. Semantic parsing is a very interesting application we plan to investigate in the future. We will include this paper in our revision.
>
> > Alg 1: What is "ancestral sampling"? Please explain.
>
> In the literature, `ancestral sampling' is commonly used to describe the default process for sampling from a locally normalized generative model, that is, sample the variables in order of a topological sort of the graphical model. In left-to-right autoregressive models like LLMs, this is equivalent to just sampling tokens left-to-right. We will add this explanation.

---

> > ### Comment · Reviewer_cHBv · 2024-08-12
> >
> > Thanks for the response. I will keep my score since I believe that some additional work (e.g. training a neural estimate of EFG) will make this paper more contained.

---

### Author Rebuttal · Authors · 2024-08-07

We thank the reviewers for their feedback, which will greatly improve the paper.

We have addressed each reviewer’s comment in their corresponding answer.

We summarize the edits we plan to do to improve the paper based on the feedback we received:
Reviewers cHBv and gp9U proposed ways to potentially modify the ASAp algorithm to speed up convergence.
1. We will clarify in introduction that the main focus of the paper is to formalize the likelihood misalignment problem in existing grammar-constrained decoding, and to provide an initial solution to address this problem together with a proof of convergence. We will explain in Section 3 that the ASAp algorithm is not necessarily optimal in terms of sample-efficiency and clarify in the conclusion that there are opportunities for improvement.
2. We will include a formal definition of CFG and an example of how a string can be derived for the CFG in Fig 1.
3. We will add the missing related work provided by the reviewer cHBv: R Shin, et al., Constrained language models yield few-shot semantic parsers (2021)
4. We will address all other comments raised by the reviewers and include all the technical and writing clarifications from each rebuttal.
5. We have performed the experiment proposed by reviewer gp9U on fine-tuned models for our tasks and did not observe any significant differences in terms of convergence rates for both GCD and ASAp compared to the original model. If the reviewers find it beneficial, we can add a paragraph about this finding.

---

### Decision · Program_Chairs · 2024-09-25

**Decision:**

Accept (poster)

**Comment:**

This paper focuses on the constrained decoding scenario where LLMs are expected to produce high-quality and grammatically correct outputs. The authors point out that prior methods on grammar-constrained decoding (GCD) distorts the language model's learned distribution over sequences. As mitigation strategy, they present the adaptive sampling with approximate expected futures (ASAp) approach, which is designed to enhance the quality of the output by adjusting the conditional probability to align with the model’s original output distribution while ensuring grammaticality. The evaluation is performed on code generation and structured NLP tasks showing a better likelihood for LLM’s outputs being grammar-constrained.

The paper is well written and the method is well-formalized and clear. The paper provides empirical validation that the method works and that it improves the benchmark scores. The main focus of the paper is to formalize the likelihood misalignment problem in existing grammar-constrained decoding which is an important finding. The notion of expected future grammatically will be impactful in the area of sequence decoding.

Unfortunately, ASAp is computationally expensive, requires building a large data structure to keep track of the sampled prefixes and many iterations. Even though this may limit the adaptation of ASAp, the main contribution of the paper is the finding that conventional methods on grammar-constrained decoding distorts the language model's learned distribution.